# Enhanced ZBTB16 Levels by Progestin-Only Contraceptives Induces Decidualization and Inflammation

**DOI:** 10.3390/ijms241310532

**Published:** 2023-06-23

**Authors:** Sefa Arlier, Umit A. Kayisli, Nihan Semerci, Asli Ozmen, Kellie Larsen, Frederick Schatz, Charles J. Lockwood, Ozlem Guzeloglu-Kayisli

**Affiliations:** Department of Obstetrics and Gynecology, Morsani College of Medicine, University of South Florida, Tampa, FL 33612, USA; sefaarlier@gmail.com (S.A.); uakayisli@usf.edu (U.A.K.); nsemerci@usf.edu (N.S.); asliozmen@usf.edu (A.O.); klarsen@usf.edu (K.L.); fschatz@usf.edu (F.S.); cjlockwood@usf.edu (C.J.L.)

**Keywords:** decidualization, endometrial stromal cells, progestin-only contraceptives, ZBTB16, IL-8, tissue factor, COX-2

## Abstract

Progestin-only long-acting reversible-contraceptive (pLARC)-exposed endometria displays decidualized human endometrial stromal cells (HESCs) and hyperdilated thin-walled fragile microvessels. The combination of fragile microvessels and enhanced tissue factor levels in decidualized HESCs generates excess thrombin, which contributes to abnormal uterine bleeding (AUB) by inducing inflammation, aberrant angiogenesis, and proteolysis. The- zinc finger and BTB domain containing 16 (ZBTB16) has been reported as an essential regulator of decidualization. Microarray studies have demonstrated that *ZBTB16* levels are induced by medroxyprogesterone acetate (MPA) and etonogestrel (ETO) in cultured HESCs. We hypothesized that pLARC-induced ZBTB16 expression contributes to HESC decidualization, whereas prolonged enhancement of ZBTB16 levels triggers an inflammatory milieu by inducing pro-inflammatory gene expression and tissue-factor-mediated thrombin generation in decidualized HESCs. Thus, ZBTB16 immunostaining was performed in paired endometria from pre- and post-depo-MPA (DMPA)-administrated women and oophorectomized guinea pigs exposed to the vehicle, estradiol (E_2_), MPA, or E_2_ + MPA. The effect of progestins including MPA, ETO, and levonorgestrel (LNG) and estradiol + MPA + cyclic-AMP (E_2_ + MPA + cAMP) on *ZBTB16* levels were measured in HESC cultures by qPCR and immunoblotting. The regulation of *ZBTB16* levels by MPA was evaluated in glucocorticoid-receptor-silenced HESC cultures. *ZBTB16* was overexpressed in cultured HESCs for 72 h followed by a ± 1 IU/mL thrombin treatment for 6 h. DMPA administration in women and MPA treatment in guinea pigs enhanced ZBTB16 immunostaining in endometrial stromal and glandular epithelial cells. The *in vitro* findings indicated that: (1) ZBTB16 levels were significantly elevated by all progestin treatments; (2) MPA exerted the greatest effect on *ZBTB16* levels; (3) MPA-induced *ZBTB16* expression was inhibited in glucocorticoid-receptor-silenced HESCs. Moreover, *ZBTB16* overexpression in HESCs significantly enhanced prolactin (*PRL*), insulin-like growth factor binding protein 1 (*IGFBP1*), and tissue factor (*F3*) levels. Thrombin-induced interleukin 8 (*IL-8)* and prostaglandin-endoperoxide synthase 2 (*PTGS2)* mRNA levels in control-vector-transfected HESCs were further increased by *ZBTB16* overexpression. In conclusion, these results supported that ZBTB16 is enhanced during decidualization, and long-term induction of ZBTB16 expression by pLARCs contributes to thrombin generation through enhancing tissue factor expression and inflammation by enhancing *IL-8* and *PTGS2* levels in decidualized HESCs.

## 1. Introduction

Progestin-only long-acting reversible contraceptives (pLARCs), e.g., injectable form of depo-medroxyprogesterone acetate (DMPA), subdermal implants releasing etonogestrel (ETO), and intrauterine devices releasing levonorgestrel (LNG), are safe and highly effective in averting unwanted pregnancies, particularly among adolescents, and preterm births by prolonging inter-pregnancy intervals [1,2]. The use of pLARCs is preferred in women in whom estrogen-containing formulations are contraindicated, e.g., during lactation or with an estrogen-dependent tumor [3,4]. Their safety, affordability, and prolonged efficacy make pLARCs particularly ideal for use in countries with limited healthcare access [5]. However, abnormal uterine bleeding (AUB) in pLARC users is the major discontinuation reason [6,7].

Progesterone withdrawal with uniform reductions in endometrial hemostasis and spiral arterial vasospasm is the cause of menstrual bleeding, while pLARC-induced AUB occurs intermittently and focally from irregularly distributed hyperdilated, superficial, and fragile microvessels [8,9]. Our prior studies in pLARC-administered women or guinea pigs showed that pLARCs reduced uterine blood flow, causing endometrial hypoxia and generating reactive oxygen species [10,11,12], which cause AUB directly by damaging endometrial microvessels and indirectly by inducing aberrant angiogenesis [13]. The microscopic evaluation of endometria from pLARC users revealed that only bleeding sites displayed abnormal microvessels enmeshed in a compromised extracellular matrix, which was contiguous with decidualized human endometrial stromal cells (HESCs). Decidualized HESCs express elevated levels of tissue factor (encoded by *F3* gene), the primary initiator of hemostasis via Factor-VIIa- and Factor-Xa/Va-generated thrombin [14,15]. Thus, pLARC-induced vascular damage generates excess thrombin by delivering circulating Factor VII to decidualized HESC-expressed tissue factor. Thrombin is well documented to prevent bleeding by activating platelets and generating fibrin. However, thrombin also displays distinct cellular effects by binding to protease-activated receptors. Thrombin: (1) paradoxically promotes endothelial permeability and focal hemorrhage [16,17,18]; (2) induces aberrant angiogenesis and inflammation by generating vascular endothelial growth factor (VEGF) [19] and interleukin–8 (IL–8) levels, respectively, in HESCs [20]; (3) enhances the expression of HESC-derived matrix metalloproteinase-1 and -3 [21,22]. Therefore, the combined actions of thrombin-mediated angiogenic, proteolytic, and inflammatory processes markedly contribute to pLARC-induced AUB [20,23].

The zinc finger and BTB domain-containing 16 (ZBTB16) protein, also known as the promyelocytic leukemia zinc finger, contains several zinc finger domains, a BTB domain, and three RD2 subdomains [24,25]. The zinc finger domains at the C-terminus bind either directly to a specific sequence in the genome or interact with other transcription factors, thereby controlling the transcriptional specificity of ZBTB16 [25]. In contrast, the BTB domain at the N-terminus mediates the formation of homo- and/or hetero-dimers with other ZBTB proteins. Therefore, the BTB domain plays a critical role in the stability, localization, and transcriptional activity of ZBTB16 [25] and is also involved in chromatin remodeling by binding to different corepressors and histone modification enzymes [26,27]. ZBTB16 is expressed by several tissues/cell types, including neuronal, muscle, hematopoietic, respiratory, and reproductive cells [28], and regulates many biological processes, including stem cell renewal, proliferation, cell cycle regulation, differentiation, and apoptosis [28,29,30,31]. ZBTB16 exerts these actions as a transcription factor and/or chromatin remodeler to repress and/or activate gene expression in a cell-type-specific or stimulation-dependent manner [32,33,34]. Previous studies showed that ZBTB16 is induced by glucocorticoid and progesterone in HESCs [35] and plays a crucial role in HESC decidualization by regulating the expression of the early growth response 1 gene [36].

Although several mechanisms are involved in the pathogenesis of AUB [13,37,38,39], a gap exists as to precisely how pLARCs trigger AUB by altering endometrial cellular and/or molecular responses. Our previous whole-genome microarray analysis identified *ZBTB16* as one of the most-highly induced genes by both medroxyprogesterone acetate (MPA) and ETO in primary cultured HESCs [40]. Thus, we hypothesized that pLARC-induced ZBTB16 expression contributes to HESC decidualization, whereas prolonged enhancement of ZBTB16 levels triggers an inflammatory milieu by inducing pro-inflammatory gene expression and tissue-factor-mediated thrombin generation in decidualized HESCs. This hypothesis was tested by investigating the in situ regulation of ZBTB16 levels in endometrial sections obtained from pLARC-administrated women or guinea pigs. Subsequently, we evaluated in vitro pLARC-mediated regulation of ZBTB16 in HESCs and human endometrial endothelial cells (HEECs), as well as the functional role of ZBTB16 in cultured HESCs to gain insight into the underlying mechanism of pLARC-induced inflammation and AUB.

## 2. Results

### 2.1. Increased Endometrial ZBTB16 Levels in Women Receiving DMPA Treatment

The immunohistochemical analysis of paired endometrial sections (n = 7) obtained from women pre- and post-DMPA administration revealed that ZBTB16 is expressed by endometrial stromal, glandular epithelial, and endothelial cells (Figure 1A). Moreover, strong ZBTB16 immunoreactivity was observed in post-DMPA-administered endometrial stromal and glandular cells, as well as superficial microvascular endothelial cells, whereas pre-DMPA endometria displayed significantly less ZBTB16 immunostaining, which was primarily localized to stromal cells (Figure 1A). HSCORE analysis (Figure 1B) confirmed significantly higher ZBTB16 levels in endometrial stromal cells (mean ± SEM; 185.4 ± 9.2 vs. 92.1 ± 13.8; *p* < 0.001), glandular cells (181.3 ± 20.4 vs. 47.7 ± 14.7; *p* < 0.001), and endothelial cells (185.3 ± 19.9 vs. 45.4 ± 16.3; *p* < 0.001) in post-DMPA vs. pre-DMPA administration. 

### 2.2. Elevated Endometrial ZBTB16 Levels in Guinea Pigs Treated with MPA

ZBTB16 immunostaining analysis of endometrial sections of guinea pigs treated 21 days with placebo (control; n = 5), or estradiol (E_2_; n = 4), or MPA (n = 5), or E_2_ + MPA (n = 6) indicated that, compared to placebo (63.6 ± 14.1): (1) neither E_2_ (48.3 ± 11.6; *p* = 0.56) nor MPA (105.0 ± 11.0; *p* = 0.11) significantly altered ZBTB16 levels; and (2) E_2_ + MPA significantly enhanced ZBTB16 immunoreactivity (128.8 ± 22.5; *p* < 0.05; Figure 2A,B) in endometrial stromal cells. In contrast, in endometrial glandular epithelial cells, significantly higher ZBTB16 expression was observed in MPA-treated endometria (153.3 ± 15.6; *p* < 0.05), but not following E_2_ + MPA (74.1 ± 18.5) or E_2_ treatment (67.0 ± 29.8) vs. the control (90.4 ± 18.6; Figure 2A,B). These results indicated differential regulation of ZBTB16 by steroids in stromal vs. epithelial cells in the guinea pig endometrium.

### 2.3. Increased ZBTB16 Levels during In Vitro Decidualization of Primary Cultured HESCs

We next evaluated *ZBTB16* mRNA levels during in vitro decidualization of HESCs treated with 10^−8^ M E_2_ + 10^−7^ M MPA + 5 × 10^−5^ M cyclic AMP (i.e., EMC) for 3 or 6 days. EMC treatment significantly increased *ZBTB16* mRNA levels in HESCs by ~200- and 400-fold at 3 and 6 days of decidualization, respectively, vs. Day 0 of induction (Figure 3A; *p* < 0.001). In addition to verifying decidualization-related morphological changes of stromal fibroblasts to epithelioid-like decidual cells, the well-documented decidualization markers, prolactin (*PRL*) and insulin-like growth factor binding protein 1 (*IGFBP1*) levels [41], were also measured by qPCR to confirm in vitro decidualization. Compared to their levels on Day 0, EMC treatment significantly induced both *PRL* and *IGFBP1* mRNA levels by up to 2-fold on Day 3 with a further increase observed for both *PRL* (by 10-fold) and *IGFBP1* (by 28-fold) during HESC decidualization on Day 6 (Appendix A; *p* < 0.05).

To determine the individual contribution of the EMC components to HESC *ZBTB16* expression, parallel experiments were performed in HESCs treated with either the control, E_2_, cAMP, or MPA for 6 days. Compared to the control, neither E_2_ nor cAMP treatment induced *ZBTB16* mRNA levels, whereas MPA treatment significantly enhanced *ZBTB16* levels by ~200-fold (Figure 3B; *p* < 0.05), confirming MPA as a primary EMC component that upregulates *ZBTB16* mRNA in HESCs.

### 2.4. In Vitro Comparison of Induction of ZBTB16 Levels by Different pLARCs

*ZBTB16* mRNA and protein levels were measured by qPCR and immunoblotting, respectively, in HESCs treated with 10^−8^ M E_2_ as the control ± various progestins used in different pLARC formulations including 10^−7^ M ETO, LNG, and the mixed progestin/glucocorticoid agonist, MPA, as well as a pure glucocorticoid, dexamethasone (DEX), and a pure progestin, Organon 2058 (ORG), for 7 days. This analysis revealed that all treatments significantly increased *ZBTB16* mRNA levels compared to the E_2_ treatment (Figure 4A; *p* < 0.05). Among these treatments, DEX was the most-effective (~280 fold), whereas ORG was the least-effective (~8-fold) inducer of *ZBTB16* levels in HESCs. However, based on the fold increases of ZBTB16 levels by individual progestins used in pLARCs, MPA was the strongest inducer, increasing *ZBTB16* mRNA levels by 150-fold, followed by ETO and LNG treatment, which increased *ZBTB16* by 47- and 15-fold, respectively (Figure 4A; *p* < 0.05). Immunoblot analysis confirmed parallel changes in ZBTB16 protein levels. Thus, compared to the E_2_-treated control, each treatment significantly elevated immunoreactive ZBTBP16 levels in HESCs, with DEX-treated HESCs displaying the greatest induction, followed by MPA, then ETO, LNG, and ORG by ~17, 9, 6, and 4-fold, respectively (Figure 4B; *p* < 0.05).

To determine if pLARC-mediated ZBTB16 upregulation is primarily glucocorticoid-receptor-dependent, HESCs were transfected with either scramble, nonspecific control siRNA, or the glucocorticoid receptor encoded by *NR3C1* gene-specific siRNA, then treated with EMC for 3 days. After confirming that the *NR3C1* mRNA levels were significantly downregulated by the *NR3C1* siRNA compared to control siRNA transfection (0.09 ± 0.01 vs. 1.02 ± 0.002; *p* < 0.001), *NR3C1*-siRNA-transfected HECSs displayed significantly reduced *ZBTB16* expression compared to the control (0.08 ± 0.01 vs. 1.03 ± 0.003; *p* < 0.001). The reduction in glucocorticoid receptor activity was confirmed by measuring the levels *of FKBP5,* which is a well-documented glucocorticoid-receptor-regulated gene (0.27 ± 0.04 vs. 1.01 ± 0.01; *p* < 0.001; Figure 4C). Thus, pLARC-mediated ZBTB16 upregulation is primarily glucocorticoid-receptor-dependent.

In addition to HESCs, cultured HEECs were also treated with E_2_ alone ± ORG, or ETO, or LNG, or MPA, or DEX to further confirm glucocorticoid-receptor-mediated ZBTB16 upregulation in endothelial cells since HEECs express the glucocorticoid receptor [42], but not the progesterone receptor [43,44]. The evaluation of *ZBTB16* levels in HEECs by qPCR showed that only MPA and DEX significantly induced *ZBTB16* mRNA levels by ~28- and 80-fold, respectively, compared to the E_2_-treated control group (Appendix A; *p* < 0.05), whereas ETO- or LNG-induced changes did not attain statistical significance, and ORG as a pure progestin did not alter *ZBTB16* expression in HEEC cultures (Appendix A).

### 2.5. ZBTB16 Overexpression Increases Expression of Decidualization Markers in HESCs

To investigate the functional role of ZBTB16 on pLARC-induced decidualization, HESCs were transiently transfected with either the *ZBTB16* expression vector or an empty vector as the control. Following transfection, the cultures were incubated with or without EMC for 3 days. As shown in Appendix A, the qPCR results confirmed that, compared to control vector transfection, the *ZBTB16* levels were significantly increased in *ZBTB16*-transfected HESCs. However, EMC did not significantly increase *ZBTB16* in transfected cells (*p =* 0.19; Appendix A).

The decidualization markers’, *PRL* and *IGFBP1*, levels were also measured in the control or *ZBTB16*-transfected HESCs. The *PRL* and *IGFBP1* levels were 1.9- and 3.4-fold higher in *ZBTB16*-vector-transfected cells vs. control-vector-transfected cells in the absence of EMC. In control-transfected cells, EMC treatment also significantly increased both *PRL* and *IGFBP1* levels by 3.4- or 2.2-fold, respectively (Figure 5A). EMC treatment further enhanced *PRL* (6.8 ± 2.2 vs. 3.4 ± 1.1; *p* < 0.05) and *IGFBP1* levels (6.0 ± 1.2 vs. 2.2 ± 0.4; *p* < 0.05) in *ZBTB16*-transfected vs. control-transfected HESCs (Figure 5A).

### 2.6. Elevated ZBTB16 in HESCs Induces Tissue Factor Expression

We previously reported that bleeding versus non-bleeding sites of pLARC-exposed endometrium display elevated tissue factor (*F3*) expression, which causes excess thrombin generation [15]. Therefore, to investigate the impact of ZBTB16 alone or in combination with thrombin on *F3* expression, cultured HESCs transfected with either the *ZBTB16* or control vector were incubated with or without EMC for 72 h, then treated with 1 U/mL of thrombin for 6 h. Subsequent qPCR analysis revealed that *F3* expression was higher in *ZBTB16*-transfected HESCs compared to control transfection (1.5 ± 0.03 vs. 1.0 ± 0.001; *p* < 0.05; Figure 5B). Moreover, thrombin significantly elevated *F3* mRNA levels in control-transfected HESCs (Figure 5B; *p* < 0.05), and thrombin-induced *F3* levels were further increased in *ZBTB16*-transfected *HESCs* vs. control-transfected HESCs treated with thrombin (2.9 ± 0.17 vs. 2.0 ± 0.04; *p* < 0.05; Figure 5, upper panel), but not in EMC-treated HESCs (Figure 5B, lower panel). Note that thrombin treatment in HESCs with or without EMC exposure did not change *ZBTB16* levels in either control- or ZBTB16-vector-transfected cells (Appendix A).

### 2.7. Elevated ZBTB16 in HESCs Promotes Inflammation by Increasing IL-8 and PTGS2 Expression

Our previous studies reported that thrombin promotes aberrant angiogenesis by enhancing VEGF expression [19] and inflammation by elevating the IL-8 level in HESCs [20]. Therefore, we investigated if ZBTB16 is involved in thrombin-mediated inflammatory processes, which may indirectly augment pLARC-induced AUB [20,23]. HESCs transfected with either the *ZBTB16* or control vector were incubated with EMC for 72 h, then treated with 1 U/mL of thrombin for 6 h. The analysis of qPCR revealed that, compared to control transfection: (1) *IL-8* and *PTGS2* (also known as cyclooxygenase 2 (COX2)) levels increased in *ZBTB16*-transfected HESCs (1.4 ± 0.1 vs. 1.0 ± 0.02 and 2.7 ± 0.1 vs. 1.0 ± 0.03, respectively; *p* < 0.05; Figure 5C); (2) thrombin significantly increased *IL-8* and *PTGS2* levels by 3.7- and 11.2-fold in control-vector-transfected cells with further induction of *IL-8* and *PTGS2* levels by 4.6- and 24.7-fold in *ZBTB16*-vector-transfected cells (Figure 5C).

## 3. Discussion

A variety of contraceptive agents including progestin-only or combined oral contraceptive pills, DMPA injections, subdermal implants, or intrauterine devices are used by women to prevent unintended pregnancies during their reproductive lifespan [45]. AUB accompanying pLARCs is a common side effect and the most-cited cause for discontinuation of these otherwise safe and effective agents. Each year, more than one-million unintended pregnancies occur because of the discontinuation or misuse of these contraceptives [46]. The bleeding sites of pLARC-treated endometria display decidualized stomal cells, as well as irregularly distributed hyperdilated thin-walled superficial vessels associated with deficiency in both basement membranes and perivascular smooth muscle cells or pericytes [8,9,39]. Clotting factor leakage from these impaired vessels interacts with the decidualized stromal cell tissue factor induced by pLARCs, thereby generating excess thrombin, which cause dysfunctional regulation of cellular pathways and/or mediators that play crucial roles in endometrial hemostasis, angiogenesis, and inflammation (Figure 6). In support of these findings, we previously showed that excess thrombin disrupts newly formed vessels [47]. The current study revealed that ZBTB16 may indirectly contribute to pLARC-associated AUB by inducing endometrial pro-inflammatory molecules *IL8* and *PTGS2*, as well as decidualization-associated *F3* levels. The integration of our in situ findings from endometria of human DMPA users and MPA-treated guinea pigs with in vitro results from *ZBTB16* overexpressing HESCs supports the potential role of enhanced ZBTB16 levels in pLARC-associated AUB.

Previous studies demonstrated that ZBTB16 is widely expressed in various tissues and displays diverse biological functions such as cell growth, self-renewal, and differentiation through the regulation of tissue-specific target genes [25,28]. In addition to the documented cellular and molecular roles of ZBTB16, in the female reproductive system, Fahnenstich et al. [35] initially reported elevated ZBTB16 expression in endometrial stromal and myometrial cells during the secretary vs. proliferative phase of the menstrual cycle, thereby revealing hormonal regulation of ZBTB16 in the human endometrium. Subsequently, Szwarc et al. [49] reported that ZBTB16 plays a crucial role during HESC decidualization by acting as a potential mediator of progesterone receptor actions. The current study expanded the potential role of ZBTB16 in the endometrium by investigating whether ZBTB16 contributes to pLARC-induced tissue-factor-mediated thrombin generation and the inflammatory milieu. Therefore, we initially evaluated ZBTB16 expression in pre- and post-DMPA-administered women and guinea pigs as a highly relevant model [9,11]. Consequentially, the results provided the first evidence that DMPA administration induces ZBTB16 expression in situ in endometrial stromal, glandular epithelial, as well as endothelial cells in both women and guinea pigs.

Decidualization represents the cellular transformation of fibroblast-like endometrial stromal cells into epithelial-like decidual cells and is essential for the establishment of a successful pregnancy [50]. Decidualization results from complex interactions of several key factors including steroid hormones, growth factors, signaling molecules, transcription factors, etc., [50,51,52]. Here, we confirmed that both morphologically and functionally differentiated decidual cells display significantly elevated *ZBTB16* levels during in vitro decidualization, as previously shown by Kommagani et al. [36]. Our investigation of the individual effect of in vitro decidualization stimuli by using E_2_, or MPA, or cAMP alone identified MPA (progesterone/glucocorticoid) as the responsible agent for the robust increase in *ZBTB16* levels in decidualized HESCs, suggesting that ZBTB16 is both a progestin- and glucocorticoid-induced transcription factor. This result was consistent with previous studies demonstrating ZBTB16 upregulation in a variety of cell types by progesterone [35], corticosteroids [53], testosterone [54], and aldosterone [55]. Taken together, these results indicate that ZBTB16 is a steroid-responsive transcription factor. 

Our current results obtained from in the situ analysis of human and guinea pig endometria, as well as in vitro HESC decidualization led us to analyze ZBTB16 expression in HESCs treated with various progestins including ORG, ETO, LNG, and MPA, which are used in pLARC formulations or DEX (a pure glucocorticoid). Although all these compounds elevated ZBTB16 levels, robust induction was observed in HESCs following either DEX or MPA exposure, confirming that glucocorticoids display a stronger effect on the upregulation of ZBTB16 mRNA and protein levels than progestins. Among progestins, ETO displayed the greatest induction in ZBTB16 levels, indicating that ETO exhibits both glucocorticoid and progestogenic effects [56]. Depending on their derivative molecules, several synthetic progestins display mixed progestin–glucocorticoid actions by binding to the progesterone and glucocorticoid receptors in target cells [57]. Previously, Kommagani et al. [36] reported that progestin-mediated ZBTB16 induction in HESCs requires a progesterone receptor. However, the results of the current study indicated that this effect is primarily mediated by the glucocorticoid receptor rather than the progesterone receptor because the MPA-induced *ZBTB16* levels were significantly diminished in glucocorticoid-receptor-silenced HESCs. This was further supported by a moderate induction of ZBTB16 by pure progestin ORG. 

Our current study also confirmed MPA-mediated ZBTB16 upregulation in HEECs, which express the glucocorticoid, but not the progesterone receptor [42,43,44]. Our findings that ZBTB16 levels were significantly upregulated in HEECs following either DEX or MPA exposure, but not by ORG, ETO, or LNG treatment, further support glucocorticoid-receptor-mediated regulation of ZBTB16 expression. Several studies reported that increased levels of ZBTB16 expression induce apoptosis and inhibit VEGF-induced proliferation, thereby acting as an anti-angiogenic factor in human umbilical cord and corneal endothelial cell cultures [58,59,60]. Thus, pLARC-induced endothelial ZBTB16 expression may impair endometrial angiogenesis to contribute to the mechanism of pLARC-induced AUB.

The current study provided the first demonstration that *ZBTB16* overexpression in cultured HESCs without decidual stimuli (EMC treatment) significantly increases the generation of tissue factor, which is specifically enhanced at bleeding sites vs. non-bleeding sites in the endometrium of pLARC users [14]. Moreover, thrombin-treated HESCs display increased levels of tissue factor, which is further elevated in *ZBTB16*-overexpressed HESCs, suggesting the mechanism whereby tissue factor elicits excess local thrombin generation, which stimulates VEGF expression, thereby exacerbating abnormal endometrial angiogenesis observed in the endometrium of pLARC users [19]. Moreover, the endometrium of pLARC users contains a higher number of neutrophils and NK cells [61,62] and displays elevated IL-8 levels [20]. In support, our results revealed that thrombin further increases *IL8* and COX2 (*PTGS2*) levels in *ZBTB16* overexpressed HESCs, indicating that elevated ZBTB16 levels promote thrombin-induced inflammation in pLARC users (Figure 6) and providing the first demonstration of the combined inflammatory effect of thrombin and ZBTB16 in HESCs. Previous studies reported that thrombin induces the expression of several cytokines and chemokines in a variety of cell types [48,63,64,65]. Moreover, ZBTB16 has been shown to display a wide range of cell-type-specific functions including cell proliferation, cell cycle regulation, apoptosis, stem cell renewal, and differentiation [66,67,68,69,70,71,72]. These wide-ranging ZBTB16-induced effects result from either an activator or repressor effect of ZBTB16 binding to either DNA-specific consensus sequences or other transcription factors, resulting in upregulation or downregulation of gene expression. Mao et al. [70] showed that ZBTB16 indirectly regulates the secretion of several cytokines such as IL-4, -13, -17, or IFNγ by binding to multiple T-helper-specific transcription factors in NKT cells. ZBTB16 can interfere with glucocorticoid-mediated transcription [71,73]. In the current study, the amplification of thrombin-induced *IL8* and *PTGS2* expression in ZBTB16-overexpressed HESCs supported the function of ZBTB16 to interfere with glucocorticoid-receptor-mediated suppression of inflammatory gene expression. However, further studies are required to verify this effect in HESCs. In summary, ZBTB16 displays diverse functions as a transcription factor and/or chromatin remodeler. Thus, it activates and/or represses target gene expression depending on physiological or pathological conditions. The current study uncovered the novel roles played by ZBTB16 by revealing that: (1) pLARC-induced ZBTB16 expression contributes to HESC decidualization; (2) increased ZBTB16 expression in the endometrium of pLARC-treated women is primarily glucocorticoid receptor mediated; (3) elevated ZBTB16 levels induce tissue factor, which elicits excess local thrombin generation, thereby contributing to AUB; and (4) thrombin treatment in ZBTB16-overexpressed HESCs further increases *PTGS2* and *IL-8* levels, thereby promoting endometrial inflammation in pLARC users. The limitation of our study is that most of our results were obtained from in vitro findings. Therefore, further in situ study is needed to test the role of ZBTB16 in pLARC-induced inflammation and AUB in higher primates.

## 4. Materials and Methods

### 4.1. Endometrial Tissue Collection

#### 4.1.1. Human Endometrial Tissue Collection

Banked paraffin sections of paired endometrial tissues obtained from women prior to DMPA injection (pre-DMPA, n = 7) and three months post-DMPA administration (n = 7) were used for immunohistochemical studies. Endometrial biopsies were obtained from women with regular menstrual cycles during the secretory phase pre-DMPA or post-DPMA administration. All samples were de-identified after receiving written informed consent at New York University under Institutional Review Board approval (#H6023) and later approval of the University of South Florida (#Pro00019480). Endometrial biopsies for primary cultures of HESCs and HEECs were obtained from healthy, reproductive-age women not receiving any hormonal/steroid treatment undergoing hysterectomy for benign disease. Endometrial tissues were collected after receiving written informed consent under the approval of the Yale University and the University of South Florida Human Investigation Committees.

#### 4.1.2. Guinea Pig Endometrial Tissue Collection

Nulliparous female guinea pigs (aged 2–6 months) after bilateral ovariectomy were administrated subcutaneous implants (based on a 21-day time release) of placebo as the control (n = 5), or 5 mg estradiol (E_2_; n = 4), or 50 mg MPA (n = 5), or in combination of 5 mg E_2_ with 50 mg MPA (n = 6) pellets purchased from Innovative Research of America (Sarasota, FL, USA). After three weeks, a hysterectomy was performed, and the right uterine horn was formally fixed for immunohistochemical studies. These studies were performed under the approval of the Yale Institutional Animal Care and Use Committee (IACUC#2006–11002).

#### 4.1.3. Immunohistochemistry

ZBTB16 immunostaining was performed on human endometrial sections obtained from women before (pre-) and post-DMPA administration, as well as endometrial tissues obtained from guinea pigs administered with placebo, E_2_, MPA, or E_2_ + MPA, as described [11]. Briefly, 5 µm-thick paraffin embedded sections were deparaffinized in xylene and rehydrated in a descending ethanol series. Antigen retrieval was performed in 10 mM citric acid solution (pH 6.0) by boiling in a microwave for 20 min. Subsequently, slides were rinsed in Tris-buffered saline with 0.1% tween-20 (TBS-T) and incubated in 3% hydrogen peroxide for 12 min at room temperature for endogenous peroxidase quenching. Thereafter, sections were blocked for 30 min at room temperature with 5% normal goat serum (Vector Labs, Burlingame, CA, USA) in TBS-T, then incubated with rabbit polyclonal anti-ZBTB16 antibody (1/150 dilution; Sigma-Aldrich, St. Louis, MO, USA) overnight at 4 °C. After the washing steps in TBS-T, the slides were incubated with biotinylated goat anti-rabbit IgG (1/400 dilution, Vector Labs) in TBS-T for 30 min, then with streptavidin-conjugated peroxidase complex (Vector Labs) for 30 min. Following several TBT-T rinses, immunoreactivity was developed using 3,3′-diaminobenzidine kits (Vector Labs), and the slides were counterstained with hematoxylin. As a negative control, non-immune rabbit IgG was used at the same concentration as the primary antibody. ZBTB16 immunoreactivity was assessed by histologic score (HSCORE) analysis, a semiquantitative method that evaluates the intensity and number of immunostained cells, as described [23], by two independent investigators (S.A., N.S.) blinded to the source of the samples. 

### 4.2. Cell Culture

Aliquots of frozen primary HESCs previously isolated and characterized, as described [40], were thawed and cultured in basal media containing Dulbecco’s MEM/F12 with 10% stripped calf serum and 1% antibiotic (Life Technologies, Carlsbad CA, USA). To induce decidualization, confluent cultures were incubated with 10^−8^ M E_2_ (Sigma-Aldrich) + 10^−7^ M MPA (Sigma-Aldrich) and 5 × 10^−5^ M cyclic AMP (Sigma; named EMC media) for 0, 3, or 6 days. In parallel, cultures were incubated in basal media containing 10^−8^ M E_2_ alone, or in combination with 10^−7^ M ORG (Merck & Co, Whitehouse Station, NJ, USA), or 10^−7^ M ETO (Sigma-Aldrich), or 10^−7^ M LNG (Sigma-Aldrich), or 10^−7^ M MPA, or 10^−7^ M DEX (Sigma-Aldrich) for 7 days. Then, cultures were switched to defined medium as described previously [23] with corresponding steroids for 6 h or 24 h for RNA and protein analysis, respectively. 

Frozen banked HEECs isolated and characterized, as described [13,74], were thawed and cultured in EGM-2 MV Medium with 5% fetal calf serum (Cambrex Bio Science, Walkersville, MD, USA). Similar to the HESCs’ treatment, confluent HEECs seeded in 6-well plates were incubated with 10^−8^ M E_2_ (as a control) ± 10^−7^ M ORG, or 10^−7^ M ETO, or 10^−7^ M LNG, or 10^−7^ M MPA, or 10^−7^ M DEX for 7 days. After serum starvation, HEECs were treated with the corresponding steroids for 6 h for RNA extraction. 

#### 4.2.1. Quantitative Real-Time PCR

Total RNA obtained from cultured HESCs or HEECs was isolated using the RNeasy mini kit according to the manufacturer’s instructions (Qiagen, Germantown, MD, USA). Then 500 ng of total RNA was used for reverse transcription, which was performed using the RETROscript kit (Ambion, Austin, TX, USA) using random decamer primers at 42 °C for 1 h, followed by 92 °C for 10 min for enzyme inactivation. The expression of target genes including *ZBTB16*, *PRL*, *IGFBP1*, *F3*, *IL8*, and *PTGS2* was determined by using a specific TaqMan gene expression assay (Life Technologies; see Appendix A). Amplification used 40 cycles of PCR in an ABI 7500 System (Applied Biosystems, Foster City, CA, USA), with the following program: initial denaturation at 95 °C for 10 min, followed by 40 cycles at 95 °C for 15 s and 60 °C for 60 s. Each reaction was performed in triplicate and the average used for each sample. The relative expression for each target gene was normalized with β-Actin as a reference gene and calculated using the comparative Ct method (2^−ΔΔCt^). 

#### 4.2.2. Immunoblotting Analyses

Total protein was isolated from cell lysates obtained from HESCs incubated with E_2_, or E_2_ + ORG, or ETO, or LNG, or MPA, or DEX for 24 h using lysis buffer (Cell Signaling Inc., Danvers, MA, USA) containing a protease inhibitor cocktail. The protein concentration was determined by a detergent-compatible protein assay (Bio-Rad Laboratories, Hercules, CA, USA). Then, 20 µg of total protein was diluted in 2× sample buffer (Bio-Rad) and then boiled for 5 min. Samples were then separated by 10% sodium dodecyl sulfate (SDS)-polyacrylamide gel electrophoresis (Bio-Rad) and transferred onto a nitrocellulose membrane (Bio-Rad). The membrane was blocked overnight with 10% non-fat dry milk in TBS-T and then incubated overnight at 4 °C with mouse anti-ZBTB16 monoclonal antibody (ThermoFisher, Waltham, MA, USA). Following washing in TBS-T, the membrane was incubated with peroxidase-conjugated anti-horse IgG at a 1/5000 dilution (Vector Labs). The signals were developed using a chemiluminescence kit (Amersham; GE Healthcare, Piscataway, NJ, USA). The membrane was then sequentially stripped and re-probed with peroxidase-conjugated anti-β-actin rabbit monoclonal antibody at 1/1000 dilution (Cell Signaling Inc.).

#### 4.2.3. Transient Transfection of ZBTB16 Overexpression

To overexpress *ZBTB16* in HESCs, the expression vector containing the human full-length *ZBTB16* gene open reading frame (pCMV6-ZBTB16) was purchased from Origene Inc. (Rockville, MD, USA). Cultured HESCs seeded in 6-well plates were transiently transfected with either the *ZBTB16* or empty (pCMV6 vector as the control) vector using the Lipofectamine LTX reagent (Invitrogen, Carlsbad, CA, USA) in Opti-MEM serum reduced medium (Invitrogen) according to the manufacturer’s instructions. After transfection for 4 h, HESCs were incubated with/without EMC media for 72 h. In a parallel set, *ZBTB16*- or control-vector-transfected HESCs cultures ± EMC were treated with 1 U/mL of thrombin (American Diagnostic, Greenwich, CT, USA) for 6 h and stored at −80 °C for RNA isolation. The transfection efficiency for *ZBTB16* overexpression was confirmed by qPCR and immunoblotting. 

#### 4.2.4. siRNA-Mediated Silencing of Glucocorticoid Receptor

To silence the expression of the glucocorticoid-receptor-encoded *NR3C1* gene, HESCs were transfected with either *NR3C1*-specific siRNA or non-specific siRNA (Invitrogen) as the control using the RNAiMax Reagent according to the manufacturer’s instructions (Invitrogen). Four hours after transfection, cultured HESCs were incubated with EMC media for 72 h. The transfection efficiency for glucocorticoid receptor silencing was confirmed by qPCR by measuring the mRNA levels of the *NR3C1* and *FKBP5* genes as a glucocorticoid-receptor-regulated gene. 

### 4.3. Statistical Analysis 

The results from ZBTB16 immunostaining in guinea pig endometria, or immunoblotting, or qPCR were each normally distributed as determined by the Kolmogorov–Smirnov test and analyzed by one-way ANOVA, followed by the post hoc Holm–Sidak method or Student–Newman–Keuls method. HSCOREs obtained from the immunostaining of human endometrial tissues from pre- and post-DMPA administration were compared using a *t*-test. *p* < 0.05 was considered statistically significant. The statistical calculations used the SigmaStat Version 3.0 software (Systat Software, San Jose, CA, USA). 

## Figures and Tables

**Figure 1 ijms-24-10532-f001:**
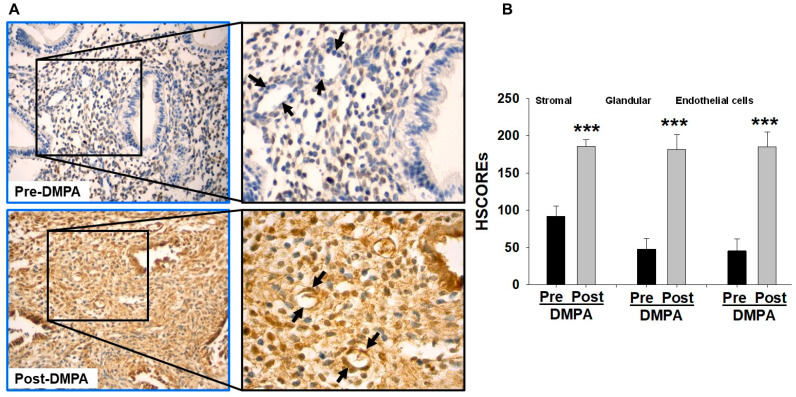
Endometrial stromal and glandular cells display enhanced ZBTB16 immunoreactivity in women administered depo-medroxyprogesterone acetate (DMPA). (**A**) Representative ZBTB16 immunostaining (brown) in paraffin sections from paired endometria obtained from pre- and 3 months post-DMPA administration. Original magnification ×40. Enhanced ZBTB16 immunoreactivity was observed in post-DMPA endometrial compared to pre-DMPA. A strong ZBTB16 immunoreactivity is seen in post-DPMA vascular endothelium (arrows) in the inset micrographs. Original magnification ×100. (**B**) HSCOREs for ZBTB16 immunoreactivity confirmed significantly higher ZBTB16 immunoreactivity in both endometrial stromal (185.4 ± 9.2 vs. 92.1 ± 13.8) and glandular cells (181.3 ± 20.4 vs. 47.7 ± 14.7), as well as endothelial cells (185.3 ± 19.9 vs. 45.4 ± 16.3; *p* < 0.001) in post- vs. pre-DMPA endometria. Bars represent the mean ± SEM; n = 7/each; *** *p* < 0.001 vs. pre-DMPA analyzed by the *t*-test.

**Figure 2 ijms-24-10532-f002:**
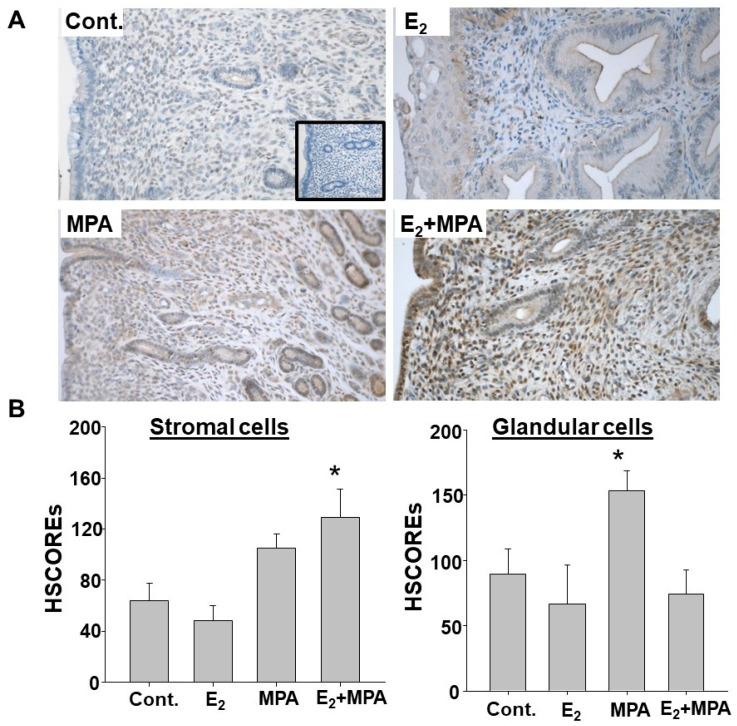
MPA administration induces endometrial ZBTB16 expression in guinea pigs. (**A**) Representative images of ZBTB16 immunostaining in endometria obtained from ovariectomized guinea pigs treated 21 days with placebo as the control (Cont; n = 5), or estradiol (E_2_; n = 4), or medroxyprogesterone acetate (MPA; n = 5), or E_2_ + MPA (n = 6). Original magnification: ×20. The insert represents negative control staining. (**B**) HSCORE analysis of ZBTB16 immunoreactivity in stromal and glandular cells of endometria. Bars represent the mean ± SEM; ** p* < 0.05 vs. the control or E_2_ treatment in stromal cells, and ** p* < 0.05 vs. the control or E_2_ treatment in glandular epithelial cells analyzed by one-way ANOVA followed by the Holm–Sidak method.

**Figure 3 ijms-24-10532-f003:**
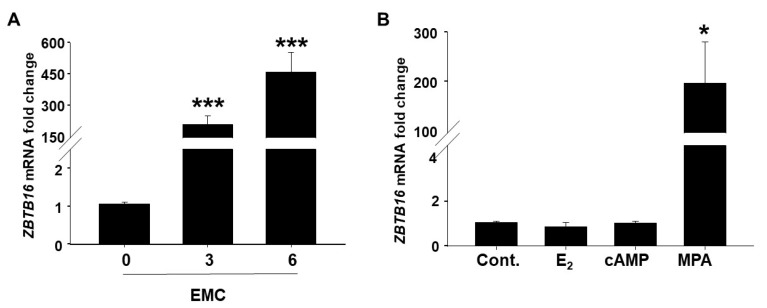
Increased *ZBTB16* expression during decidualization of cultured HESCs. (**A**) *ZBTB16* mRNA levels in HESCs treated with 10^−8^ M E_2_ + 10^−7^ MPA + 5 × 10^−5^ cyclic AMP (EMC) for 0, 3, or 6 days by qPCR. The data represent the fold change as the mean ± SEM; n = 5/each; *** *p* < 0.001 vs. Day 0 analyzed by one-way ANOVA followed by the Student–Newman–Keuls method. (**B**) MPA mediated upregulation of *ZBTB16* mRNA levels in HESCs treated with vehicle (control; Cont) or E_2_ or MPA or cAMP for 6 days. Bars represent mean ± SEM; n = 4/each; * *p* < 0.05 vs. Cont or E_2_ or cAMP analyzed by one way ANOVA followed by Student-Newman-Keuls method.

**Figure 4 ijms-24-10532-f004:**
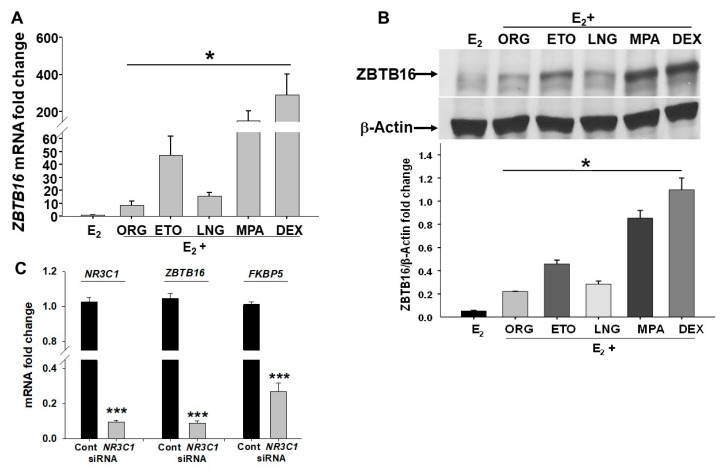
pLARCs induce ZBTB16 mRNA and protein levels in cultured HESCs. *ZBTB16* mRNA (**A**) and protein (**B**) levels were analyzed by qPCR and immunoblotting, respectively, in HESCs treated with 10^−8^ M E_2_ ± 10^−7^ M ORG, ETO, LNG, MPA, or DEX for 7 days. Bars represent the mean ± SEM; (**A**) n = 4 for mRNA fold change; * *p* < 0.05 vs. E_2_ alone; and (**B**) n = 3 for protein levels after normalization to β-actin; * *p* < 0.05 vs. E_2_ alone analyzed by one-way ANOVA followed by the Student–Newman–Keuls method. E_2_: estradiol, ORG: Organon 2058, ETO: etonogestrel, LNG: levonorgestrel, MPA: medroxyprogesterone acetate, DEX: dexamethasone. (**C**) *NR3C1*, *ZBTB16,* and *FKBP5* mRNA levels in HESCs transfected with either the nonspecific control (Cont) or *NR3C1*-specific siRNA and treated with EMC for 3 days. Bars represent the mean ± SEM; n = 3; *** *p* < 0.001 vs. control siRNA by the *t*-test.

**Figure 5 ijms-24-10532-f005:**
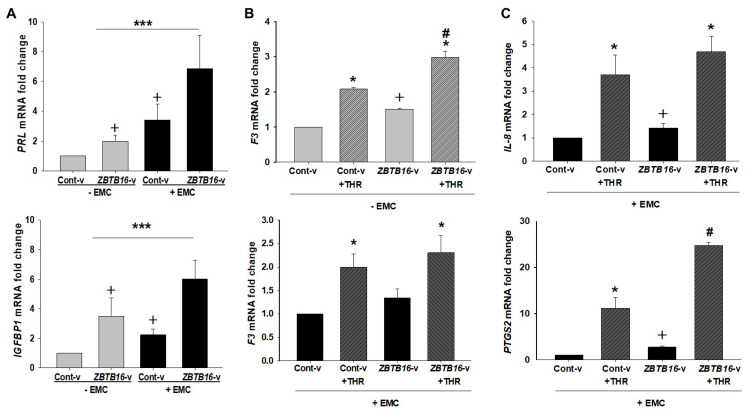
*ZBTB16* overexpression induces decidualization markers, as well as *F3*, *IL-8*, and *PTGS2* levels in HESCs. (**A**) Decidualization markers’, *PRL* and *IGFBP1*, mRNA levels in *ZBTB16-*vector (*ZBTB16*-v) or control-vector (Cont-v)-transfected HESCs treated with or without 10^−8^ M estradiol + 10^−7^ M medroxyprogesterone acetate + 5 × 10^−5^ M cAMP (EMC) for 3 days. Bars represent the mean ± SEM; n = 4; *** *p* < 0.05 vs. control-v-EMC; ^+^
*p* < 0.05 vs. *ZBTB16*-v + EMC. (**B**) Tissue factor (*F3*) mRNA levels in control- (Cont-v) or *ZBTB16*-vector (*ZBTB16*-v)-transfected HESCs treated with or without EMC for 3 days ± 1 U/mL of thrombin (THR) for 6 h. Bars represent the mean ± SEM; n = 4; * *p* < 0.05 vs. Cont-v or *ZBTB16*-v; ^+^
*p* < 0.05 vs. Cont-v; ^#^
*p* < 0.05 vs. Cont-v + THR. (**C**) Interleukin 8 (*IL-8*) and prostaglandin endoperoxide synthase 2 (*PTGS2 aka* cyclooxygenase 2; COX2) mRNA levels in the control- or *ZBTB16*-vector-transfected HESCs treated with EMC for 3 days ± 1 U/mL THR for 6 h. Bars represent the mean ± SEM; n = 4; * *p* < 0.05 vs. Cont-v or *ZBTB16*-v; ^+^ *p* < 0.05 vs. Cont-v; ^#^
*p* < 0.05 vs. Cont-v + THR. The data were analyzed by one-way ANOVA followed by the Student–Newman–Keuls method.

**Figure 6 ijms-24-10532-f006:**
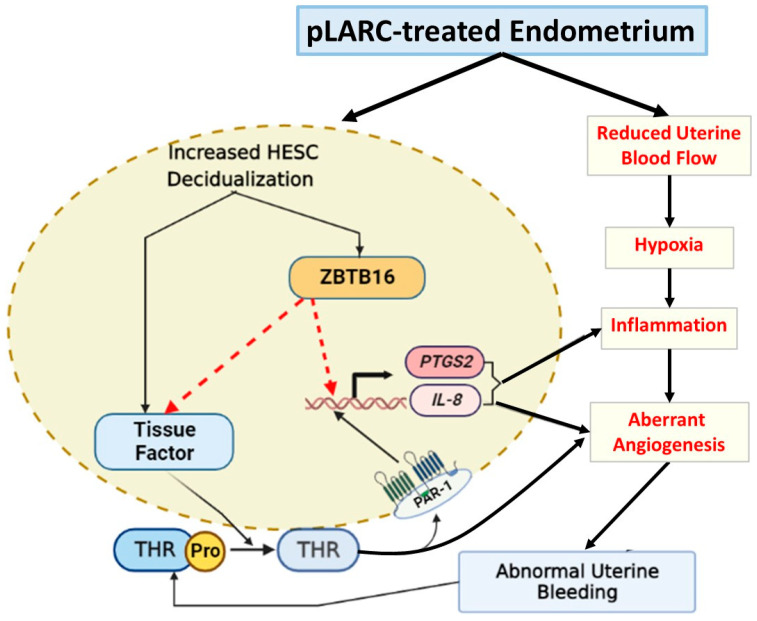
Schematic demonstration of the role of elevated ZBTB16 expression in the pathogenesis of pLARC-induced AUB. Administration of progestin-only, long-acting, reversible contraception (pLARCs) reduce uterine blood flow, which causes local hypoxia [8,9] and induces HESC decidualization, which increases tissue factor [15,23] and ZBTB16 levels. Increased ZBTB16 levels induce excess tissue factor expression, which generates thrombin. The resulting excess thrombin binds to protease-activated receptors (PARs), which increase the expression of several inflammatory factors and angiogenic factors, such as VEGF or IL-8 [19,20,48]. These factors promote excess angiogenesis, which results in abnormal uterine bleeding. pLARC-induced excess ZBTB16 exacerbates thrombin-induced endometrial angiogenesis and inflammation by increasing *IL8* and *PTGS2* levels in HESCs.

## Data Availability

Data supporting the results of the current study are available from the corresponding author upon request.

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
