# Peer review of "Enhanced ZBTB16 Levels by Progestin-Only Contraceptives Induces Decidualization and Inflammation"

_ijms, 2023, doi:10.3390/ijms241310532_

Round 1
Reviewer 1 Report
IJMS
COMMENTS TO THE EDITORS AND THE AUTHORS
ijms-2366590: ”Progestin-Only Contraceptives Induce Decidualization by Enhancing ZBTB16 Expression:Implications for Abnormal Uterine Bleeding”
Dear the Editors and the Authors,
Please find enclosed the comments for the above-mentioned manuscript.
A SUMMARY OF THE CONTENT
The authors hypothesized that progestin-only long-acting reversible contraceptive (pLARC) induction of ZBTB16 is involved in the pathogenesis of pLARC-associated abnormal uterine bleeding. Results showed involvement of ZBTB16 in progestin-only contraceptives induced decidualization. The pLARCs-induced ZBTB16 expression contributes to HESC decidualization. Increased ZBTB16 expression in the endometrium of pLARC treated women is primarily glucocorticoid receptor mediated. Elevated ZBTB16 levels induce tissue factor and local thrombin generation. Thrombin treatment in ZBTB16 overexpressed HESCs further increases PTGS2 and IL-8 levels.
THE OVERALL OPINION OF THE MANUSCRIPT
The strengths: the manuscript is within the scope of the journal and perfectly fits to the special issue subject; the results present new and interesting knowledge; the results were obtained using the different methods; most of the figures very nicely present results.
The limitations: the title and the conclusions are not fully supported by the data since direct connection of the presented results with abnormal uterine bleeding is missing; Figure 6 is not supported by the results; the intra- and inter- assay coefficients are not presented.
Please find some of the suggestions in the comments to the authors listed below.
(1) TITLE
Please consider rewriting the title to precisely reflect the results presented in the manuscript since the direct connection of the presented results with abnormal uterine bleeding is not shown.
(1a) KEY WORDS
1a.1. Please consider removing “abnormal uterine bleeding” since the direct connection of the presented results with abnormal uterine bleeding is not shown.
1a.2. Please consider adding some of the molecular markers affected with ZBTB16 expression and/or manipulation.
(2) ABSTRACT
2.1. Please rewrite the hypothesis since the presented results do not show the direct connection with abnormal uterine bleeding.
2.2. Please rewrite the entire text of the abstract in order to better present results. Namely, just 1/3 of the text of the abstract are results. The results related to the increased endometrial ZBTB16 level in women receiving DMPA treatment and the elevated endometrial ZBTB16 level in guinea pigs treated with MPA are missing.
2.3. Please make text related to the very well-known fact shorter.
2.4. Please avoid using “abnormal uterine bleeding” since the direct connection of the presented results with abnormal uterine bleeding is not shown.
(3) INTRODUCTION
Lines 98-100: “Thus, we hypothesized that pLARC-98 induced ZBTB16 expression contributes to HESC decidualization, and that prolonged enhancement of ZBTB16 levels promote AUB by inducing excess thrombin generated by decidualized HESC expressed tissue factor.”
Please consider rewriting the text since direct connection of the presented results with abnormal uterine bleeding is not shown.
(4) MATERIALS AND METHODS
4.1. Please describe methods in more details.
4.2. Please provide the amount of RNA used for RQ-PCR.
4.3. Please provide the amount of proteins used for WB.
4.4. Please provide intra- as well as inter-assay coefficients.
(5) RESULTS
Please avoid using “abnormal uterine bleeding” in the text related to the presented results since the direct connection of the presented results with abnormal uterine bleeding is not shown.
(6) DISCUSSION
6.1. Please discuss the original and important pioneered results, as well as recent advance in the field focusing on the subject of the study.
6.2. Please avoid using “abnormal uterine bleeding” in the text related to the presented results since the direct connection of the presented results with abnormal uterine bleeding is not shown.
6.3. Please discuss the limitation of the study since most of the results were obtained from in vitro experiments.
(7) REFERENCES
7.1. Please provide references describing the original (not reviews), and important and pioneered results, but also references describing the recent advance in the field.
(8) FIGURES and FIGURE LEGENDS
Please modify Figure 6 to reflect presented results or precisely state/decipher the presented results from the results obtained by other authors.
(9) SUPPLEMENTARY MATERIALS
Lins502-503. The titles are missing.
(10) GENERAL
Please avoid using “abnormal uterine bleeding” in the text of the manuscript related to the presented results since the direct connection of the presented results with abnormal uterine bleeding is not shown.
Good luck and all the best!
Author Response
The authors hypothesized that progestin-only long-acting reversible contraceptive (pLARC) induction of ZBTB16 is involved in the pathogenesis of pLARC-associated abnormal uterine bleeding. Results showed involvement of ZBTB16 in progestin-only contraceptives induced decidualization. The pLARCs-induced ZBTB16 expression contributes to HESC decidualization. Increased ZBTB16 expression in the endometrium of pLARC treated women is primarily glucocorticoid receptor mediated. Elevated ZBTB16 levels induce tissue factor and local thrombin generation. Thrombin treatment in ZBTB16 overexpressed HESCs further increases PTGS2 and IL-8 levels. The strengths: the manuscript is within the scope of the journal and perfectly fits to the special issue subject; the results present new and interesting knowledge; the results were obtained using the different methods; most of the figures very nicely present results. The limitations: the title and the conclusions are not fully supported by the data since direct connection of the presented results with abnormal uterine bleeding is missing; Figure 6 is not supported by the results; the intra- and inter- assay coefficients are not presented.
Title R#1.1. Please consider rewriting the title to precisely reflect the results presented in the manuscript since the direct connection of the presented results with abnormal uterine bleeding is not shown.
Response: As suggested by the Reviewer, the title of the manuscript is revised as “Enhanced ZBTB16 Levels by Progestin-Only Contraceptives Induces Decidualization and Inflammation.”
Keywords: R#1.1a.1. Please consider removing “abnormal uterine bleeding” since the direct connection of the presented results with abnormal uterine bleeding is not shown.
Response: “Abnormal uterine bleeding” is now deleted in the Keywords section.
Keywords: R#1.1a.2. Please consider adding some of the molecular markers affected with ZBTB16 expression and/or manipulation.
Response: As molecular markers, “IL-8, Tissue Factor and COX-2” are now included in the Keywords section.
Abstract R#1.2.1. Please rewrite the hypothesis since the presented results do not show the direct connection with abnormal uterine bleeding.
Response: As suggested by the Reviewer, we revised the hypothesis as “pLARC-induced ZBTB16 expression contributes to HESC decidualization, whereas prolonged enhancement of ZBTB16 levels triggers an inflammatory milieu by inducing pro-inflammatory gene expression and tissue factor-mediated thrombin generation in decidualized HESCs.”
Abstract R#1.2.2. Please rewrite the entire text of the abstract in order to better present results. Namely, just 1/3 of the text of the abstract are results. The results related to the increased endometrial ZBTB16 level in women receiving DMPA treatment and the elevated endometrial ZBTB16 level in guinea pigs treated with MPA are missing.
Response: The statement indicating, “enhanced endometrial ZBTB16 levels in women and guinea pigs” was presented in lines 24-25 of the original submission (Yellow highlighted sentence ion the revised manuscript). We also revised material and method, results as well as conclusion sections as suggested.
Abstract R#1.2.3 Please make text related to the very well-known fact shorter.
Response: Abstract section is significantly shortened according to the Reviewer’s suggestion.
Abstract R#1.2.4. Please avoid using “abnormal uterine bleeding” since the direct connection of the presented results with abnormal uterine bleeding is not shown.
Response: We deleted “abnormal uterine bleeding” and revised the sentence as “long-term induction of ZBTB16 expression by pLARCs contributes to thrombin generation through enhancing tissue factor expression and inflammation by enhancing IL-8 and PTGS2 levels in decidualized HESCs.”
Introduction R#1.3. Lines 98-100: “Thus, we hypothesized that pLARC-induced ZBTB16 expression contributes to HESC decidualization, and that prolonged enhancement of ZBTB16 levels promote AUB by inducing excess thrombin generated by decidualized HESC expressed tissue factor.” Please consider rewriting the text since direct connection of the presented results with abnormal uterine bleeding is not shown.
Response: As suggested by the Reviewer, we revised the hypothesis as “pLARC-induced ZBTB16 expression contributes to HESC decidualization, whereas prolonged enhancement of ZBTB16 levels triggers an inflammatory milieu by inducing pro-inflammatory gene expression and tissue factor-mediated thrombin generation in decidualized HESCs.” (Page 3, 2nd paragraph, lines 11-12).
M and M R#1.4.1. Please describe methods in more details.
Response: As suggested, more details are included in immunohistochemistry, immunoblotting, qPCR sections in the revised manuscript.
M and M R#1.4.2. Please provide the amount of RNA used for RQ-PCR.
Response: The statement “500 ng total RNA” is now included into the revised manuscript.
M and M R#1.4.3. Please provide the amount of proteins used for WB.
Response: The statement “20 µg of total protein” is now included into the revised manuscript.
M and M R#1.4.4. Please provide intra- as well as inter-assay coefficients.
Response: Average inter- or intra-assay CVs percentage for qPCR experiments were less than 5%. Endometrial sections from human or guinea pigs were immunostained on same bench and inter-assay coefficients were less than 10% for two independent investigators (SA, NS). For ZBTB16 immunoblotting, all protein lysates from cultured HESCs (n=3) were loaded one time and primary/secondary antibody as well as chemiluminescence development were performed together in the same loading. Thus, there are no inter- or intra-assay coefficients.
Results R#1.5. Please avoid using “abnormal uterine bleeding” in the text related to the presented results since the direct connection of the presented results with abnormal uterine bleeding is not shown.
Response: As suggested by the Reviewer, we revised the sentences expressing direct connection between ZBTB16 and abnormal uterine bleeding throughout the revised manuscript, which includes the following sentences from the Results section.
Page 8 2nd paragraph Line 1: “To investigate the potential role of ZBTB16 on pLARC-induced decidualization…”
Page 9 Line 4: “2.6. Elevated ZBTB16 in HESCs Induces Tissue Factor Expression”
Page 9 Line 21-22: “…if ZBTB16 is involved in thrombin-mediated inflammatory processes, which may indirectly augment pLARC-induced AUB [20, 23].”
Discussion R#1.6.1. Please discuss the original and important pioneered results, as well as recent advance in the field focusing on the subject of the study.
Response: In the revised version, additional studies are also included.
Discussion R#1.6.1.6.2. Please avoid using “abnormal uterine bleeding” in the text related to the presented results since the direct connection of the presented results with abnormal uterine bleeding is not shown.
Response: As responded for R#1.5, the statements pointing out direct connection between ZBTB16 and AUB are revised in the discussion of the revised manuscript.
Page 11 Lines 6-11: The current study reveals that ZBTB16 may indirectly contribute to pLARC-associated AUB by inducing endometrial pro-inflammatory molecules IL8 and PTGS2 as well as decidualization-associated tissue factor levels. Integration of our in-situ findings from endometria of human DMPA users and MPA-treated guinea pigs with in vitro results from ZBTB16 overexpressing HESCs supports the potential role of enhanced ZBTB16 levels in pLARC-associated AUB.
Page 12 Lines 11-13: The current study expands the potential role of ZBTB16 in the endometrium by investigating whether ZBTB16 contributes to pLARC-induced tissue factor-mediated thrombin generation and inflammatory milieu.
Discussion R#1.6.1. Please discuss the limitation of the study since most of the results were obtained from in vitro experiments.
Response: As suggested by the Reviewer, we added the following statement “A limitation of our study is that most of our results were obtained from in vitro findings. Therefore, further in situ study is needed to test the role of ZBTB16 on pLARC-induced inflammation and AUB in higher primates.” Page 14 Lines 2-5.
Reference R#1.6.1.7.1. Please provide references describing the original (not reviews), and important and pioneered results, but also references describing the recent advance in the field.
Response: Thirteen new references from research studies are included by replacing with review articles in the revised manuscript.
Results R#1.6.1. Please modify Figure 6 to reflect presented results or precisely state/decipher the presented results from the results obtained by other authors.
Response: The potential mechanisms represented in the Figure 6 and legends are obtained from our previous studies. However, all findings are now cited by appropriate references in the Figure 6 legend.
Suppl Mat R#1.6.1.Line 502-503. The titles are missing.
Response: We added the title and legend for Supp Figure 1.
Reviewer 2 Report
The authors of the presented study show effect of progestin only contraceptives on inducing ZBTB16 expression and its possible implications in abnormal uterine bleeding. They used gain-of-function studies and gene silencing experiments to elucidate the involvement of the gene in decidualization, its action through glucocorticoid receptor, and its role in the induction of tissue factor. Moreover they show that ZBTB16 in association with thrombin increases expression of pro-inflammatory IL-8 and PTGS2.
The study presents some interesting results regarding possible mechanisms of pLARC induced AUB through ZBTB16. However, the studies would have benefited from an experiment showing effect of thrombin on PAR-1 (as depicted in schematic) in ZBTB16 over-expressed cells.
Major concern:
How do Authors reconcile the findings of their in vivo and in vitro results concerning effect of MPA on ZBTB16 expression in stromal cells. In Figure 2, MPA does not show any significant effect on stromal cells, while all the in vitro experiments were performed using stromal cells.
Why didn't the authors evaluate direct effect of ZBTB16 on HEECs? In my opinion it would have strengthened the studies as a direct effect on apoptosis/angiogenesis using HEECs would provide a more direct evidence for AUB.
Author Response
Responses to Comments by Reviewer 2:
The authors of the presented study show effect of progestin only contraceptives on inducing ZBTB16 expression and its possible implications in abnormal uterine bleeding. They used gain-of-function studies and gene silencing experiments to elucidate the involvement of the gene in decidualization, its action through glucocorticoid receptor, and its role in the induction of tissue factor. Moreover, they show that ZBTB16 in association with thrombin increases expression of pro-inflammatory IL-8 and PTGS2.The study presents some interesting results regarding possible mechanisms of pLARC induced AUB through ZBTB16. However, the studies would have benefited from an experiment showing effect of thrombin on PAR-1 (as depicted in schematic) in ZBTB16 over-expressed cells.
Response: According to the Reviewer’s suggestion, to investigate whether the impact of ZBTB16 on PAR1 expression, the levels of PAR1 (F2R1 gene) were analyzed by qPCR in cultured HECS transfected with either control or ZBTB16 expression vector and treated with thrombin for 6 h. However, we did not find any significant changes in PAR1 levels in ZBTB16 overexpressed cells, consistent with its constitutive expression in HESCs, suggesting that ZBTB16 does not directly regulate PAR1 gene expression. Moreover, we and others have previously shown that thrombin by binding to PAR1 induces IL-8 expression (Lockwood et al. J Clin Endocrinol Metab. 2004:1467-75; Mhatre et al. Am J Reprod Immunol. 2016: 29-37; Kawano et al. Hum Reprod. 2011: 407-13).
R#2. How do Authors reconcile the findings of their in vivo and in vitro results concerning effect of MPA on ZBTB16 expression in stromal cells. In Figure 2, MPA does not show any significant effect on stromal cells, while all the in vitro experiments were performed using stromal cells.
Response: We agree with the Reviewer that MPA alone induces ZBTB16 levels, partially in ovariectomized guinea pigs. However, significantly enhanced ZBTB16 immunostaining was found in endometrial stromal cells of guinea pigs treated with E2+MPA, which represents a more accurate model of the physiological endometrial environment of pLARC treated women.
Why didn't the authors evaluate direct effect of ZBTB16 on HEECs? In my opinion it would have strengthened the studies as a direct effect on apoptosis/angiogenesis using HEECs would provide a more direct evidence for AUB.
Response: In this study, we primarily aimed to understand the functional role of ZBTB16 on endometrial stromal cells. We agreed with the Reviewer that enhanced ZBTB16 levels in vascular endothelium and HEECs treated with MPA and DEX might have significant contribution to pLARC-induced AUB. Our investigation of the direct effect and inhibition of ZBTB16 on endometrial endothelial cells is in progress.
Round 2
Reviewer 2 Report
I am satisfied with the overall revisions